# The Development of Equestrian Policies in China between 2015 and 2020

**DOI:** 10.3390/ani12151913

**Published:** 2022-07-27

**Authors:** Jiaxin Li, Enrique López Adán, Alfonso de la Rubia Riaza

**Affiliations:** Facultad de Ciencias de la Actividad Física y del Deporte(INEF), Universidad Politécnica de Madrid, 28040 Madrid, Spain; enrique.lopez@upm.es (E.L.A.); alfonso.delarubia@upm.es (A.d.l.R.R.)

**Keywords:** horse activities, equestrian, China, sport policies

## Abstract

**Simple Summary:**

One of the most important industries in China’s history has been the horse industry, and the modern horse industry is still in the process of being transformed. The purpose of this paper is to introduce relevant policies through an analysis of information related to the horse industry in order to promote the economic development of the horse industry and gradually create a chain for the horse industry by selecting for this analysis the fastest-growing period in China’s modern horse industry. This can support the economic development of the horse industry and gradually create an integrated horse industry with Chinese characteristics. Equestrian education in China should be supported so that it gradually develops Chinese characteristics, and, thus, horse breeding and welfare will also improve in the future.

**Abstract:**

China competed in equestrian sports for the first time at the 2008 Beijing Olympics. China’s modern equine business has developed significantly over the past decade, and the lessons from the expansion between 2015 and 2020 are important considerations as China implements further legislation to aid in the development of its modern equestrian sports. Equestrian sports can propel the Chinese horse industry forward, and the horse industry is a unique business in that it integrates one, two, and three industries, with much growth potential. This paper assesses the development of equestrianism in China from four perspectives: the general sports economic environment; the development of equestrian activities; the economic industries driven by equestrian activities; and relevant equestrian policies. Equestrian sports within China are currently facing problems, and recommendations are given. This paper is a single case study. The research utilized a qualitative approach, collecting data from official and semi-official documents. Through understanding the data collated and its analysis, equestrian sports can improve the speed and quality of their development under the influence of better-informed policy-making and a relevant economic model. It is expected that the wider related horse industry in China will also undergo more significant development.

## 1. Introduction

Local horse activities in China have a long history. China’s domestic horses were involved in establishing the foundations for commerce, communication, and state infrastructure along the old Silk Road, while also driving significant military, social, and political developments in the country [1]. As far back as the Yuan Dynasty (1271), horse activities were practiced in Beijing; after the Mongolian capital was established there, people brought their favorite horse activities to Beijing, which gradually became popular [2].

Traditional Chinese horse activities in the Ming Dynasty (1368) were still practiced, especially in spring. With the demise of the Ming Dynasty, the Manchus controlled Beijing and the Qing Dynasty (1636) arrived. During the Qianlong period (1736–1796), many race courses appeared in Beijing. Traditional Chinese horse racing events were held during various folk festivals; all levels of society, from nobles from the royal family down to the public, were enthusiastic about this activity [3]. The activity was promoted by the royal family and gradually spread to the people; this lasted until the early Republic of China (1912–1949) [4]. Around the end of the Qing Dynasty’s Xuan Tong era (1909–1911), an event called “horse racing” emerged in Beijing, which was not traditional Chinese horse racing, but so-called “Western-style horse racing” [5]. This “horse racing” activity was practiced intermittently until the late Republic of China. After the founding of the People’s Republic of China, equestrian sports gained attention again, and in the late 1950s, the state decided to launch equestrian sports nationwide. In 1979, the China Equestrian Association was established, and in 1982 it joined the International Equestrian Federation, which since 1983 has presided over national equestrian competitions and three Olympic disciplines (dressage, show jumping, and three-day event) as well as folk equestrian sports.

Since the 29th Modern Summer Olympic Games were held in Beijing in 2008, equestrian sports have been brought to the attention of the Chinese public. A notable event was the relocation of the equestrian competition venue to Hong Kong. The reason for this was the lack of an FEI-certified disease-free zone, which meant that mainland China was in an epidemic zone for equine infectious diseases [6]. Since there are no qualified disease-free zones, high-level riders from abroad can only buy one-way tickets for their horses to fly to mainland China. In many countries, the health authorities do not consider it safe to let those horses return home. Because of this, many riders choose to not compete in mainland China. This was one of the problems that made it difficult to develop equestrian sports. After many efforts to fight for and continuously improve the work, Conghua City was established as the first certified temporary certificated epidemic-free zone for international equestrian competitions in mainland China for the Asian Games 2010. This was the prelude to equestrian competition in China [7].

Policies promoting national fitness and the growth of the sports sector have been announced in recent years, particularly during the 13th Five-Year Plan era. Table 1 summarizes these policies. To further accelerate the growth of the sports industry, in 2014, the State Council released the Opinions of the State Council on Accelerating the Development of the Sports Industry and Promoting Sports Consumption; since the sports industry has gained the attention of the whole society, equestrianism also received policy support. Along with the rapid growth of equestrian clubs, there has been a successive introduction of policies related to the sports industry and the increase in demand for equestrian sports. In 2016, China announced the National Fitness Plan (2016–2020), which clearly states that there are many sports with consumer-led characteristics, including equestrian sports; the 13th Five-Year Plan for Sports Development and the Health China 2030 plan also state this. The State Council released the Notice on Issuing the Outline for the Construction of a Strong Sports Country in 2019, stating that sports would become a significant business by 2035. In 2035, the sports sector will become the foundation of the national economy. China continued to issue policies in 2021, such as the Implementation Plan for the National Fitness Facilities Project during the 14th Five-Year Plan and the National Fitness Plan (2021–2025) (after this, referred to as the Plan) to promote the development of the sports industry and achieve the goal of comprehensive fitness. Notably, in this year’s plan, but also in the 14th Five-Year period of national fitness, the total size of the national sports sector will reach RMB 5 trillion by 2025 [8]. This shows China’s determination to develop its sports industry, and with implementation of policies, sports will develop, including this niche sport. Various companies are entering the equestrian market, which means that when the market becomes more competitive, prices will go down, resulting in more people participating.

In the 1990s, China wanted to speed up the development of equine sports while exploring the economy after the reform and opening up. Most people set their sights on horse racing, which could bring more benefits, but all such projects were called off in 1999, and since then, horse racing venues that were built have been abandoned. Athletic equestrian sports then began to be developed. Following the 2008 Olympic Games, China saw a surge in equestrian fever. Since the State Council’s Opinions on Accelerating the Development of the Sports Sector and Promoting Sports Consumption (No. 46) were released in late 2014, the sports industry garnered widespread public attention, while equestrianism also gained official backing. When document No. 46 [9] was released in late 2014, the sports business was formally recognized as a distinct category in China’s economy. The report clearly states that, by 2025, a reasonably laid out, functional, and complete sports industry system will be established, the role of other industries to drive it will be significantly increased, and the total size of the sports industry will be more than RMB 5 trillion [8]. This will increase the number of equestrian clubs and participants in China. Numerous equestrian clubs have popped up, and the presence of Chinese participants at major international horse shows and auctions has increased.

In 2018, the State Council’s General Office issued the Guidelines on Accelerating the Development of Sports Competition and Performance Industry, which states that “developing the sports competition and performance industry is critical for tapping and unleashing consumption potential, safeguarding and improving people’s livelihoods, and reviving economic growth.” This strategy paved the way for the growth of equestrian sports. In 2020, China’s Ministry of Agriculture and Rural Affairs and the State General Administration of Sports released the National Equine Industry Development Plan (2020–2025), which for the first time combines horses and sports in order to plan the equine industry’s future development. This plan’s primary objective is to encourage equine breeding via athletic events, especially equestrian events, cultural tourism, and other means. By 2025, the framework and system for developing China’s modern equine industry should be established. The industry’s economic and social benefits will be significantly enhanced, with a relatively complete system of modern breeding and horse welfare that will comprise efficient breeding, treatment, and health care; disease prevention and control; training, performance measurement, competition, and performance; fitness and leisure; cultural tourism; and manufacturing of products.

Immediately after that, several municipalities in China implemented various measures to support and promote equestrian sports. Still, there were no official statistics to track the growth of equestrianism in China until 2016, by which time it had gradually gained public awareness. The purpose of this paper is to compile and analyze the development factors of equestrian sports in China based on the actual data, focusing on the period 2015 to 2020. It also points out the existing problems and offers suggestions for better development of the equestrian industry in China in the future.

## 2. Methods and Data Analysis

In this research, the single case study was predicated on looking for congruence and corroboration as well as assessing the importance of published sport-related policies and documentation [10]. We utilized a qualitative approach, collecting data from official and semi-official documents. A broad range of keywords was used in our search in order to include a wide variety of sectors (such as the equestrian industry and specialized and equestrian events; in Chinese, these are: “马术运动,” “马匹,” “马术,” “马”) from 2015 to 2020. Because horse racing is prohibited in China, it was not discussed. The data for this study came from three sources. The sources ranged from official annual reports of relevant sports authorities and the local government to academic publications and newspapers, as well as internet sources, including the State Council, the State General Administration of Sports, the Ministry of Finance and National Development and Reform Commission jointly, the Department of Science and Education, the State General Administration of Sports, the General Office of the State Council, and the National Bureau of Statistics’ *China Statistical Yearbook*, which includes information on the size of the sports industry from 2015 to 2019 and the number of horses imported to China between 2016 and 2020. Data were also collected from semi-official mass media resources, newspapers, two popular Chinese equestrian magazines in China (*Horsemanship* and *World Equestrian*), the China Physical Education Practitioner Training Network, Equestrian online, Guangming Daily news, daluma.com, JMedia, Xinhua, and government publications (National Sports Strategy). Finally, data were also collected from other yearbooks published by various provincial and municipal governments.

This analysis excludes Taiwan, Hong Kong, and Macau. Additionally, this paper refers to all provinces, municipalities, and autonomous areas in China as provinces. The years shown in the charts differ depending on the data collected. This study makes no interpretation of the facts presented. Through a graphical analysis, data from the sources above were utilized to accomplish the study goals.

### Data Analysis

Thematic analysis was used to process the collected qualitative data [11]. The work of Ryan and Bernard assisted in defining the themes of the study through research questions and theoretical frameworks (defined periods), including the equestrian political context, financial support, sports industry development, and equestrian market achievements [12]. Our team verified all parts of the themes describing equestrian development in China.

## 3. Results

### 3.1. Chinese Economy during 2015–2019

Equestrian sports are not low-cost. With the increase in income for Chinese people, there has been a rise in the number of equestrian participants. According to the data from 2015 to 2019 (Table 2), there was a considerable increase. In the first half of 2021, GDP was RMB 5,321,167 billion, up 12.7% from the first half of the previous year, or an average of 5.3% for the last two years. Disposable income for ordinary Chinese citizens increased by 12.0% in real terms, with a two-year average real annual rate of 5.2% [13]. With the fast growth of China’s economy, the amount of material goods, constant socioeconomic development, and expanded spare time all contribute to a rise in people’s need for leisure and amusement. By 2020, there will be 3.713 million sporting venues nationally, covering 3.10 billion m^2^, or 2.20 m^2^ per inhabitant. The percentage of people aged 7 and older who regularly engage in sports and exercise is about 37.2% throughout the year [13].

Data from China’s National Bureau of Statistics show that since 2015, the total output (total scale) of the national sports industry was RMB 1.7 trillion, with an added value of RMB 549.4 billion, accounting for 0.8% of the GDP in the same period; RMB 1.9 trillion in 2016, with an added value of RMB 647.5 billion, accounting for 0.9% of the GDP in the same period; and RMB 2.2 trillion in 2017, with an added value of RMB 781.1 billion. In terms of growth, the total output in 2017 was 15.7% higher than in 2016. The added value increased by 20.6%. In 2018, the total scale (total output) of the national sports industry was RMB 265.79 billion, the added value was RMB 100.78 billion, and the proportion of added value to the GDP reached 1.1%. In 2019, the total scale (total output) was RMB 2948.3 billion, and the added value was RMB 1124.8 billion. In terms of nominal growth, total output increased by 10.9%, and added value increased by 11.6% compared to 2018. Since 2018, the National Bureau of Statistics has included sports management activities; sports competition and performance activities; sports fitness and leisure activities; sports stadium services; sports intermediary services; sports training and education; sports media and information services; other sports-related services; when sports goods and related products manufacturing are combined with the sports service industry for overall statistics, we found that the size of China’s sports industry has grown year by year, from RMB 1.7 trillion in 2015 to RMB 2948.3 billion in 2019, an increase of nearly 70%. Figure 1 describes the enormous changes in the national sports industry from 2015 to 2019 in detail.

### 3.2. Sports Industry Development in China

China has 11 cities with a population of more than 10 million, more than 100 cities with a population of more than 5 million, and more than 200 cities with a population of more than 2 million. In the US, large cities such as New York and Los Angeles promote the growth of professional sports in a way no other nation can. Competitive and professional sports have origins in the cities where they are played.

In China, a new economic growth plan is based on a “dual circulation” [15] economy, in which the domestic sector would dominate and the internal and external sectors would complement each other [16]. A significant portion of the service sector is non-tradable in the dual circulation strategy. The development level of the service sector is favorably associated with a country’s economic development and national income, of which the sports industry is an important part [17].

As shown in Figure 2, the China Equestrian Association reports that the country’s equestrian industry grew from RMB 90.9 billion in 2016 to RMB 138.3 billion in 2019, representing a 51% increase. Moreover, the market capitalization of this industry has been steady every year. The 2019 equestrian training market size of RMB 13.83 billion in 2019 equates to a national penetration rate of 0.72% (3.52% in major first-tier cities). It is estimated that if a national penetration rate of 3% is achieved, the industry size will reach RMB 58 billion; if a penetration rate of 6% is achieved, the industry size will reach RMB 115 billion [18].

With the growth of the sports industry from 2015 to 2019, the rise in benefits every year makes it easier for the industry to grow. With an average growth rate of about 18%, China’s sports industry grew from 1.35 trillion to 2.2 trillion between 2014 and 2017. It was RMB 404 billion in 2014 and RMB 780 billion in 2017, with an average annual growth rate of 19%. The added value of the sports industry was 404 billion yuan in 2014. In 2014, sports output was 0.64% of GDP. By 2017, that figure had risen to 0.94%. Based on the data, it appears that growth in the overall sports industry, as well as in various industries and sports within the industry, fueled an increase in the value of the equestrian market, from RMB 9.09 billion in 2016 to RMB 13.83 billion in 2019, an increase of 52.1% in four years, with an average annual growth rate of over 10%.

### 3.3. Department of Chinese Equestrian Activities

#### 3.3.1. Social Environment

The Shijingshan Country Equestrian Club was founded in 1989, and became the first club in China to introduce competitive technology in sport. Such clubs now have many coaches and riders who are still perceived as being prestigious, as they were in prior times. With China’s rise in equestrian activities comes the development of equestrianism in the country. As shown in Figure 3, according to *Horsemanship* magazine in 2015, the number of equestrian clubs in China grew steadily after the 2008 Olympics in Beijing, from approximately 200 in early 2010 to more than 2000 in early 2019. In the last decade, there has been a 26% compound annual growth rate. According to the latest statistics from China’s *Equestrian* magazine, as of August 31, 2019, out of 2272 equestrian clubs in China, 112 were closed and 2160 were open [20]. From the data on class cost, we found that the business items of externally operated equestrian clubs are mainly annual membership cards, horse boarding, lesson fees, and sales of tack supplies. The average annual membership fee in East China is the highest, with an average of RMB 24,032/year. Shanghai has the highest average annual membership fee of RMB 38,403/year. The statistics show that the average horse boarding fee per horse is RMB 41,281/year, the average club lesson fee is RMB 488/session, and the Shanghai Club membership fee is up to RMB 38,403 per year [20]. On the social media side, professional media outlets such as internet portals have grown in popularity, along with the equestrian and horse sector. With the advancement of information technology, many marketing companies’ and institutions’ own video channels have become vital portals for equestrian media. According to TikTok data, equestrian events are the 4th most popular Olympic event [21].

It appears that there are approximately 420,000 equestrian club members throughout China, according to *Horsemanship* magazine’s survey in 2015 [19]. Club membership is dominated by teenagers, with children and teens accounting for 77% of the total, a significant increase from 2018, as shown in Table 3. On the other hand, adult members have dropped dramatically, now making up just 23% of the total. The market for teenagers has grown significantly during the last three years.

In equestrian sports, people and animals compete side by side, and the fact that men and women can compete on an equal footing over a broad age range indicates a level of equality seldom seen in other sports [22]. We evaluated the ratio of male to female members and discovered that the number of male members has been declining while the number of female members has been growing for three years. Based on the data in Table 4, club membership is still female-dominated.

#### 3.3.2. Equestrian Education

Currently, China does not have a local teaching system, and most clubs use mature foreign equestrian teaching systems. Table 5 shows a summary of equestrian teaching systems for these years in China, and Figure 3 shows the increases in clubs. It can be seen that 90% of students were taught by BHS, whereas just 9% were introduced to other methods in 2017. After 2018, many big clubs in China began to use new educational methods, such as the French GALOP system, the German FN system, the Australian Pony Club system, the Belgian VLP system, and the Dutch KNHS system. Among them, the British BHS system is still used the most, accounting for 75.97% by 2019 (2160 clubs), but there was a significant decline from 85.81% in 2018. The French Galop system accounted for 6.76% in 2018, up 15.58% that year.

During the project launch ceremony for the 2019 China Sports Education Practitioner Training Center mass equestrian sports in Beijing, equestrian sports officially became a sport and project for the masses, and the industry itself experienced a breakthrough and improvement [23].

#### 3.3.3. Horse-Breeding-Related Policies

The essential part of any equestrian sport is the horse. Equestrian activities require the breeding of high-quality horses, which necessitates a long selection, maturation, and testing process [24]. According to the 2019 equestrian industry research report shown in Figure 4, the number of imported warmbloods has decreased year-on-year since 2017, falling roughly 40% from 500 in 2015 to 300 in 2018 (the numbers are for the previous year). Meanwhile, the number of thoroughbreds increased by 28% in 2018, the number of imported Akhal horses decreased from 180 in 2015 to 30, and the numbers of miniature, Mongolian, and Orlov horses increased. By estimating the proportions of imported horses, we discovered that warmbloods, thoroughbreds, and miniature horses accounted for a significant portion of the total.

According to the published policies, five initiatives are underway to build a modern horse-breeding system. First, fundamental principles involve insisting on market orientation, responding to the need to upgrade the consumer structure, taking a leading role in emerging markets such as equestrian sports, equine cultural activities, and horseback riding tours, and enhancing the synergy of multiple sectors of horse services. Second, there are three focused tasks: a combination of equestrian sports promotion, popularization, and improvement; active promotion of youth equestrian sports; and the promotion of traditional and national equestrian activities. Third, there will be more tournament events, horse-related tourism, and integrated industrial integration support, including the following:Enhanced information sharing and mutual recognition in the industry, gradually forming a complete information management system of sport horse breeding, registration, conditioning, performance measurement, training, auction, competition, retirement, etc., and promotion of organic integration of all links in the industry chain.Exploration of horse culture resources for the purpose of creating unique local characteristics, promoting the national horse culture, and enhancing the involvement of the masses.Development of thematic and characteristic horseback riding tours, and promotion of the combination of the horse industry and tourism.Indicators of the amount of breeding and development of horses and planning for the tournament level and the synergistic effect of all segments of the industry.Equine breeding and sports event planning based on regional needs.

### 3.4. Future Development Route of Equestrian Sports

Additionally, the strategy outlines three components for the future development of equestrian sports. First, one must integrate promotion, popularization, and improvement. China should make the best possible use of the platform for Olympics preparations to promote and popularize equestrian sports. Arranging a series of tournaments and training events may help raise the competition level. Vibrant equestrian events should be organized to increase public knowledge of equestrian sports. Second, one must aggressively promote juvenile equestrian sports. Teenagers are the future of equestrian sports in China, and the country should make more effort to integrate equestrianism into schools so that more young people acquire an interest. Third, one must encourage traditional and indigenous equestrian sports. One must investigate the unique events associated with certain locations and horse breeds in China, and promote, publicize, and enhance the country’s traditional equestrian events. One must combine the features of Chinese horses, ensure good performance in special horse events, and contribute to the growth of equestrian sports on a global scale.

## 4. Discussion

Why is horse breeding important for equestrian sports? In most countries, horse racing drives the development of horse breeding, but in China, horse racing is forbidden, so horse breeding relies on the equestrian sports. There are currently only 15 common native breeds in China [25]. The country’s indigenous horse breed resources comprise ten breeds that are threatened with extinction [26]. Following from the old horse industry, a new industry, including primary, secondary, and tertiary sectors, is gradually forming, with a significant concentration on recreational riding. Difficulty arises because equestrian activities focused on high-end riding require a significant number of sport horses, a need that China’s indigenous horses cannot meet. Chinese horse breeding and development are still in the early stages, so horses with low character, speed, and power are produced.

On the other hand, many counties have a standardized horse registration system, rigorous stallion screening, and breed improvement strategies. This has resulted in the present state of equestrian contests, with higher levels of training relying heavily on imported horses. This is confirmed by a study by the Chinese Equestrian Association. According to a survey report on the development status of China’s equestrian industry and expert estimates, China has imported around 2000 horses every year from 2015 to the present, mainly from the Netherlands, New Zealand, Australia, France, and the United States. An expert study indicates that imported horses are costly, ranging from several hundred thousand to several million yuan, and the cost of shipping, inspection, and quarantine for each horse can exceed RMB 100,000. Every year, more than one billion yuan is spent on importing 2000 horses from other countries [19]. That is why strengthening China’s horse breeding is the only option.

Regarding the social aspect, from the growth of the industry in recent years, there are more equestrian clubs, and more teenagers are joining equestrian training, but there has not been a major jump in how they are taught. An annual production value of more than RMB 10 billion, an equestrian population of more than one million, more than 100 national equestrian events and activities, 1000 regional events, and 2000 equestrian clubs reflect the economic performance. At present, the China Equestrian Association only has a rider certification test, and there is no systematic teaching part. Education systems from other countries are still used, and China’s education system has not yet been formed. We also examined equestrian policies from 2016 to 2021, and we found that in the current national economy, equestrian sports, which are costly, are becoming more popular with the public. This means more people will join the sport. It is even better when the sport combines one, two, or three different industries, because that means it can grow even faster.

From the equestrian-related data of Germany, in the past 40 years, the number of horses doubled, and more than 300,000 individuals make a living directly or indirectly from horseback riding and equestrian activities [27]. In Germany, 2.32 million people call themselves riders, there are about 1.25 million privately owned horses, and the horse industry is estimated at EUR 6.7 billion [28].

In China, 2019 was the first year when statistics were available to report. There are 2160 registered clubs, with an average of 316 members per club, among which 72.96% have full-time horse workers, with an average of 8 per club, and 18.59% have foreign instructors [20]. From this, it can be deduced that the total number of Chinese riders in 2019 was 682,560. In comparison with the enormous market demand, the growth of the equestrian industry has clear flaws. The national horse population dropped from 11 million in the 1980s, when it was the greatest in the world, to around 3.6 million now. This means that there is wasting of horse resources in China. The horse industry is one of the few in the world that can combine the primary, secondary, and tertiary sectors, with a lengthy industrial chain and high volume, therefore increasing domestic demand and creating jobs for a substantial number of employees [29].

However, breeding horses involves hazards and lengthy time periods, and results will not be seen immediately. Breeding horses for sport and military reasons was viewed as an English gentleman’s patriotic obligation in the 19th century, and the creation of the English thoroughbred was probably dependent on the importation of foundation stock by English nobility. In nineteenth-century sport, horse breeding also served as a means of entry into the upper echelons of British society; however, it was a high-risk endeavor [30]. The risks associated with breeding sport horses entail a significant financial investment. As a result of the low return, horse breeding is seen as a hobby rather than a business. The horse-racing industry, which could promote horse breeding, is banned in mainland China, so horse breeding is limited to only the cultivation of FEI-certified equestrian sport. A lack of support from the horse-racing industry will result in slow horse breeding, higher sport horse prices, and more expensive equestrian sports, thereby making it difficult within the economic cycle to develop the entire horse industry.

In contrast, the cost of horses used for competition is exceptionally high, implying that the cost of equestrianism as a sport is also extremely high. With the Chinese government’s plan to increase horse breeding, it can be predicted that a more vigorous breeding program will increase the number of horses and propel the growth of the entire industry. Equestrian sports in China will also benefit from a better breeding program, as it will reduce the costs for the sport, which will attract more people to participate.

The current horse market in China is already firmly established due to several policies promoted by the government, and because equestrian clubs, equipment, and competitions have progressively improved over the past five years. From its presence in three Olympic Games in 2008 to the more recent breakthrough, Chinese equestrianism has been supported at the national and industrial level, as well as by numerous equestrian organizations.

## 5. Conclusions

We conclude that government support, the economic situation of the sports sector, and the current situation of equestrian development are the most critical aspects in developing the equestrian industry and gradually gaining popularity. More and more young people are participating in equestrian sports. With the Chinese government introducing various policies to support the development of the horse industry, the horse breeding industry will also improve and gradually form a virtuous cycle.

The equestrian industry’s rise is due to the repeated implementation of sports-related legislation and the increase in consumer demand for equestrian activities. Policy encouragement, combined with the rapid growth of China’s middle-income population in recent years, has fueled the continuous upgrading of sports consumption, and the demand for middle- and high-end sports such as equestrian sports, which was previously considered a niche, is now also increasing. In the future, we think equestrian sports in China will be popular and localized.

### Limitations

For our study, we gathered the majority of data from the government and big news organizations, and there is a lack of data from prominent equestrian clubs. Because few people have conducted in-depth research on the industry driven by the sport of equestrianism, we hope that this paper provides an overview of the horse industry, and the value of this industry will hopefully draw people’s attention, so that more people in the future will pay attention to this modern industry that combines horses and sport.

## Figures and Tables

**Figure 1 animals-12-01913-f001:**
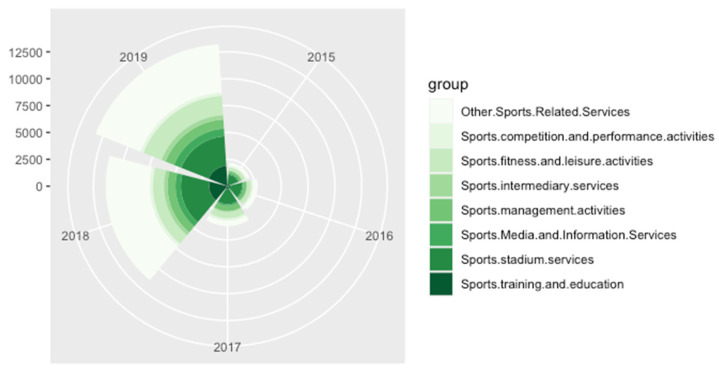
National sports industry of China from 2015 to 2019, data from [14].

**Figure 2 animals-12-01913-f002:**
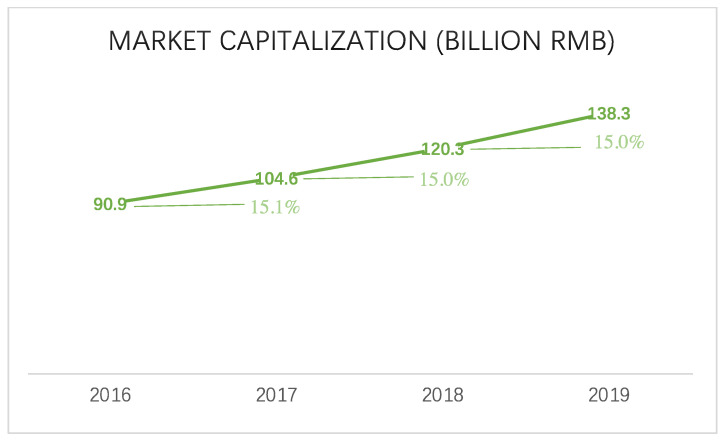
Equestrian market (billion RMB), data from [19].

**Figure 3 animals-12-01913-f003:**
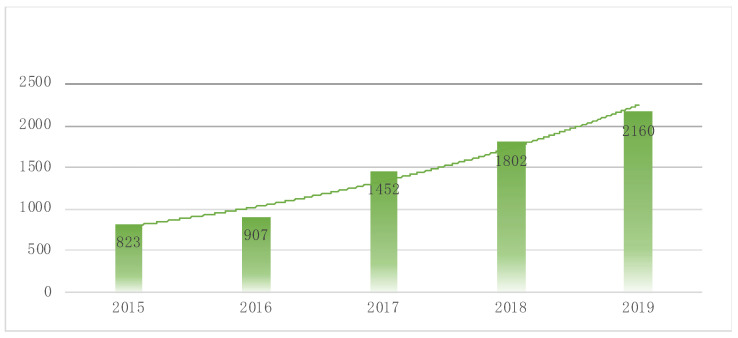
The number of equestrian clubs in China, data from [19].

**Figure 4 animals-12-01913-f004:**
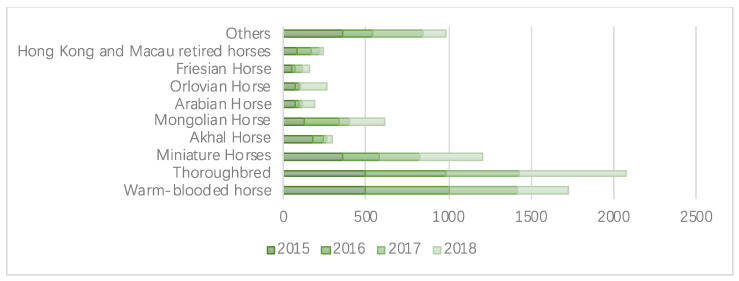
The proportion of imported horse breeds, data from [19].

**Table 1 animals-12-01913-t001:** Overview of China’s equestrian policies [8].

Equestrian Industry Policies of China			
Policy	Date	Posting Department	Keywords
Several Opinions of the State Council on Accelerating the Development of Sports Industry and Promoting Sports Consumption	2 October 2014	State Council	Fitness and leisure, particular sports, equestrian
Several Opinions of the General Administration of Sports on Promoting the Reform of the Approval System of Sports Events	24 December 2014	State General Administration of Sports	Decentralization, cancellation of approval, social forces
Notice on Issues Relating to the Cancellation and Suspension of Several Administrative Fees	29 September 2015	Ministry of Finance, National Development and Reform Commission jointly	Decentralization, cancellation of approval, social forces, equestrian competition, horse quarantine
Notice of the Department of Science and Education of the General Administration of Sports on Matters Relating to the Exemption of Outstanding Athletes from Entering Higher Education Institutions in 2016	3 November 2015	Department of Science and Education, State General Administration of Sports	Exemption from examination, equestrian athletes, higher education institutions
Notice of the State Council on the Issuance of the National Fitness Plan (2016–2020)	15 June 2016	General Office of the State Council	Comprehensive fitness, fashion and leisure, equestrian, consumer-led features, network, amateur sports
The Guidance of the General Office of the State Council on Accelerating the Development of the Fitness and Leisure Industry	25 October 2016	General Office of the State Council	Featured sports, equestrian, youth, insurance, land policy, new forms of sports media, national economy

**Table 2 animals-12-01913-t002:** Sports economics of China; data from [14].

Classification	2015				2016				2017				2018				2019			
	Economic Output (100 Million RMB)		Structure (%)		Economic Output (100 Million RMB)		Structure (%)		Economic Output (100 Million RMB)		Structure (%)		Economic Output (100 Million RMB)		Structure (%)		Economic Output (100 Million RMB)		Structure (%)	
	Gross Output	Output Growth	Gross Output	Output Growth	Gross Output	Output Growth	Gross Output	Output Growth	Gross Output	Output Growth	Gross Output	Output Growth	Gross Output	Output Growth	Gross Output	Output Growth	Gross Output	Output Growth	Gross Output	Output Growth
National sports industry	17,107	5,494.4	100	100	19,011.3	6,474.8	100	100	21,987.7	7,811.4	100	100	26,579	10,078	100	100	29,483.4	11,248.1	100	100
Sports service industry	-	-	-	-	-	-	-	-	-	-	-	-	12,732	6,530	47.9	64.8	14,929.5	7,615.1	50.6	67.7
Sporting goods and related products sales, trade agencies, and rentals	3,508.3	1,562.4	20.5	28.4	4,019.6	2,138.7	21.1	33	4,295.2	2,615.8	19.5	33.5	13,201	3,399	49.7	33.7	13,614.1	3,421	46.2	30.4
Construction of sports venues and facilities	155.2	35.3	0.9	0.6	222.1	50.3	1.2	0.8	459.6	97.8	2.1	1.3	646	150	2.4	1.5	939.8	211.9	3.2	1.9

**Table 3 animals-12-01913-t003:** Population distribution, data from [19].

Year	Children	Teenagers	Adults
2017	14.97%	46.86%	38.17%
2018	17%	49%	34%
2019	20%	57%	23%

**Table 4 animals-12-01913-t004:** Gender proportion, data from [19].

Year	Gender	Sampling Proportion
2017	Male	37.30%
	Female	62.70%
2018	Male	34.46%
	Female	65.54%
2019	Male	34%
	Female	66%

**Table 5 animals-12-01913-t005:** Changes in the teaching system, data from [19].

Year	BHS	Galop	FN	KNHS	Pony
2017	91%	9%			
2018	85.81%	6.76%	6.76%	0.67%	-
2019	75.97%	15.58%	4.55%	3.25%	4.55%

## Data Availability

The data presented in this study are available on request from the corresponding author.

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
