# Peer review of "The Development of Equestrian Policies in China between 2015 and 2020"

_animals, 2022, doi:10.3390/ani12151913_

Round 1

Reviewer 1 Report

I appreciate all the work that went into this study. However, the authors could improve some aspects.

In the text they don't differentiate between sport horses under policies of the FEI and race horse that follow other regulations (Probably the jockey club). The equine industry can be very different regarding the laws under it plays. Moreover, the monetary issues around the race horses are biggest than sport horses such us dressage or show jumper. Races are more a business than equestrian sport to be include in the number of equestrian clubs. Thoroughbred are the predominant breed, so I assume that races are the more predominant modality of ride. It is a very bias example compare the German industry (based in sport horses for dressage and jumping) with the chinesse one (predominantly of racing). 

Line 150-151: There are currently only 15 common native horses in China. I hope the authors refers to 15 native breeds not to 15 horses. 

Reviewer 2 Report

I am familiar with many aspects of this Industry and this study has considerable merit. However as it is qualitative it is vital that the information provided  and then analysed in absolutely clear.

I was not able to properly assess this paper because the use of English is unclear. This recognises the challenges for those for whom it is not the first language.

To illustrate this I attach my revision of what I think the main body of the paper says in the summary and abstract. If my version is correct, then it shows revision is needed, but if not, it also needs revision as that is what the main body said to an English speaker.

May I suggest, from previous experience as an editor, that you ask a colleague within the University with fluency in English  to review that manuscript section by section, you all working together side-by-side in real time: The authors summarise what they are trying to say in their native languages, the English speaker advise whether they read the output of  manuscript in the same way. I hope this suggestion helps and proves an interesting and indeed revealing exercise.

I would then be very happy to re-assess this interesting  paper

Author Response

Thank you for your letter and for the reviewers’ comments concerning our manuscript entitled  “The influence of the development of horse-related(Equestrian) policies in China between 2015 and 2020” (ID: animals-1828600). We appreciate the constructive comments on our manuscript, as they provide valuable guidance for revising and improving it, as well as the importance of guiding our research in the future. We have studied the comments carefully and have made a correction which we hope meets with approval. Revised portions are marked with different colors on the manuscript. Thank you for revising our summary and abstract. The main revisions in the manuscript and the responses to the reviewer’s comments are as follows:

First, we improved the English throughout the manuscript, hopefully making the language more precise and clear. Secondly, detailed changes were made to address the content to make it more logical and coherent.

Since there are fewer research articles available for equestrian sports in China, we would appreciate it if you have any recommendations for references.

Once again, thank you very much for your constructive comments and suggestions, which would help us both in English and in-depth to improve the quality of the manuscript.

Kind regards.

Round 2

Reviewer 2 Report

See editors comments

Author Response

Dear reviewer, thanks for your suggestion. We feel sorry for our English writing. Thank you for your understanding and for giving us chances to make it better. This time, we used MDPI English editing to help improve our manuscript, and we hope the revised manuscript could be acceptable to you.